# Perceptions of Caregivers Regarding Malnutrition in Children under Five in Rural Areas, South Africa [note 1]

**DOI:** 10.3390/children9111784

**Published:** 2022-11-21

**Authors:** Thabathi T. Eldah, Maluleke Mary, Raliphaswa N. Selina, Masutha T. Cecilia

**Affiliations:** Department of Advanced Nursing Science, University of Venda, Private Bag X 5050, Thohoyandou 0950, South Africa

**Keywords:** caregivers, children under-five, malnutrition, perception

## Abstract

Children under five depend on the caregivers to provide them with nutritious food to combat malnutrition. Several studies have been conducted about malnutrition in children, yet limited attention has been paid to the caregivers. Thus, the study investigated the perceptions of caregivers with regard to non-nutritious foods in rural areas in South Africa. This study explored caregiver’s perception regarding non-nutritious food in children below five. This was done in primary health care clinics of Tshilwavhusiku local areas of Makhado Municipality in Vhembe District, South Africa. A qualitative approach was adopted. Nine caregivers were sampled randomly. In-depth individual interviews were conducted, and Tesch’s analytical approach was adopted to analyze the data measures to ensure trustworthiness and ethical considerations were adhered to throughout the study. The study revealed that caregivers were lacking knowledge regarding nutritious food to be given to their children and signs of malnutrition were also not known. It is imperative to ensure the improvement of caregivers’ knowledge regarding nutritious food and children’s nutritional status in order to reduce the malnutrition rate.

## 1. Introduction

According to the Children’s Act 2005 [1], children under five are defined as people between 0 and 59 months. The Bill of Rights in the South African Constitution Section 28(1) (c) guarantees every under-five child the right to basic nutrition [2]. Nutrition plays an essential role in children’s growth, development, health, psychosocial functioning, and productivity [3,4]. Good nutrition for children includes the right amounts of vitamins, minerals, carbohydrates, protein, and fats. Breast milk is recommended as nutritious food in children below six months of age as it contains carbohydrates, protein, and fats [5]. The complementary foods for children between six and 59 months should contain different types of food groups, i.e., starchy foods, protein, vegetables, fats, and oils, and they should drink 6–8 glasses of clean, safe water [6].

All children under five depend on caregivers to have sufficient nutritious food [7]. Caregivers are responsible for providing under-five children with nutritious food daily since under-five children cannot do that themselves. This is in line with children’s act 38 of 2005, which defines caregivers as a person 16 years old and above, besides a parent or guardian, who can take care of the under-five children. The South African government supports children in need with the Child Support Grant (CSG), which is meant to buy nutritious food [8].

However, malnutrition in children under five is still alarming. Globally, among children under the age of five, 157 million are stunted, 41 million are overweight, and 50 million are wasted [9]. Africa also has a high prevalence of stunting, wasting, and being overweight. In 2019, two out of five stunted children and one quarter of the world’s wasted and overweight children under the age of five years lived in Africa [9]. According to the South Africa Demographic and Health Survey (SADHS) of 2016, wasting was found to be 2.5%, underweight at 6%, and stunting was at 27% amongst children under the age of five years [10]. According to DHIS (District Health Information System), 24 of 111 (21, 6%) and 24 of 141 (17%) children in the Kutama Clinic of the Vhembe District were reported to experience malnutrition between 2014 to 2016 and 2018 to 2019 respectively.

Several studies have been conducted about malnutrition in children under five years. Cumber et al. [3] conducted a study about poor complementary feeding practices among young children in Cameroon. Murad et al. [11] investigated the understanding of factors affecting breastfeeding practices in one city in the Kingdom of Saudi Arabia. Mohseni et al. [12] focused on factors associated with nutrition among under-five-year-old children in Iran. Omotoye et al. [13] focused on Infant and young child- feeding practices in two local government areas in southwest Nigeria. Mushaphi et al. [14] conducted a study about recommendations for infant feeding policy and programs in Dzimauli region, South Africa. Motadi et al. [15] focused on breastfeeding knowledge and practices among mothers of children younger than two years from a rural area in the Limpopo Province, South Africa.

Malnutrition in Makhado municipality was a great concern. It occurred in the same child or same household in 13 out of 24 children in 2014 to 2016. This prompted the researcher to investigate the perceptions of caregivers regarding malnutrition in children under the age of five years in rural areas in South Africa.

## 2. Materials and Methods

### 2.1. Research Design

This paper adopted a qualitative approach. An explorative, descriptive, and contextual design was also adopted. The approach was suitable for this paper, since the caregivers explored their perceptions regarding malnutrition which could not be captured in numbers, and the participants narrated their lived experiences at their home which is their contextual setting.

### 2.2. Study Setting

The study was conducted in Primary Health Care clinics of Tshilwavhusiku local area of the Makhado Municipality in Vhembe District, Limpopo Province. Out of 115 fixed primary health care clinics in the Vhembe district, the study focused on only three clinics with a high rate of child malnutrition out of 6 clinics and one health center.

### 2.3. Study Population 

The population of this study included caregivers of children who are malnourished and are under the age of five years. 

### 2.4. Sampling Method

Sampling techniques were purposive and only caregivers with malnourished children were selected as they will be able to give full and rich information needed by the researcher. Three top clinics (Kutama, Tshilwavhusiku, and Madombidzha) out of seven in the Tshilwavhusiku local area with a high rate of malnutrition in children under five years of age were sampled purposively. The DHIS (district health information system) of Vhembe was the source of the top three clinics. Caregivers was selected purposively. From the list of all children with malnutrition in each clinic, Venda-speaking participants only with a full physical address and contact details were selected.

### 2.5. Sample Size

The researcher expected to sample 5 caregivers per clinic for a total of 15 for 3 selected clinics. However, data saturated at the 9th participant and the researcher continued until the 15th to ensure that there was no new information coming out. Their information was not included as it was a repetition of what had been already said by the first 9 participants.

### 2.6. Data Collection 

Trustworthiness was ensured by applying the framework of [16]. The participants were visited by the researcher prior to the interviews to build rapport and trust. Appointments were made with the participants telephonically prior to the interviews by the researchers to tell them about the study. Then, caregivers’ homes were visited to give them full information about the study, and to give them the opportunity to talk about their perceptions regarding malnutrition in children under five, as they are people with relevant information, which could assist the researchers to achieve the aim and objective of the study. The caregivers consented verbally to be part of the interview. Date and time that was convenient to them for the interview was scheduled.

Data were collected through in-depth individual interviews at the participant’s homes at times favorable to caregivers. The open-ended interview was conducted in Tshivenda, guided by one central question: *“***Can you kindly share with me what you think may be contributing to malnutrition in the child?**” Unstructured interviews are usually qualitative in nature, and can be very helpful for social science or humanities research focusing on personal experiences. Furthermore, participants are able to express their experiences and feelings freely. Each interview lasted for approximately 30–45 min. Effective communication skills were used: active listening to encourage the participants to speak more, reflection for participants to elaborate more, and clarification on statements not understood and this helped the participants to be at ease and give more information until no more new information was coming out from the interviews researcher to ensure credibility. A tape recorder was used throughout the interview as permitted by the participants to capture all information given to participants. Furthermore, at the end of each interview, the tape recorder was played back for the participants to verify the recorded information and to identify gaps, as a way to ensure member check. The purpose of the member check was to get rid of researcher bias when interpreting the results. The researcher withheld from influencing the participant’s findings to ensure conformability. Research methods were fully described to ensure transferability. Transferability was obtained through a dense description of research methods. The research findings were also discussed with the participants. Data were transcribed verbatim.

### 2.7. Data Analysis

Data were analyzed by a team of researchers using Tesch’s approach by Creswell [17]. The voice recorder was played and recorded information transcribed verbatim. The researchers read through the transcribed data several times to get the meaning of the responses from caregivers. Then, they arranged the data into subcategories and categories, labelled them using the actual words and language of the caregivers, and came up with the theme, which appears to be the major findings of the study. 

The relation between the subcategories and categories was cross-checked to see if they relate to what caregivers said during interviews, referring to the recorder. The findings of the study were then interpreted; explaining what has been learned about caregivers’ perceptions regarding malnutrition in children under the age of five. The interpretation of the findings was grounded by the researcher’s understanding regarding malnutrition, with the integration of various kinds of literature regarding malnutrition.

### 2.8. Ethical Considerations

Prior to starting the research process, permission to conduct the study was obtained from the University of Venda Research Ethics Committee (SHS/18/PDC/05/1505), The Limpopo Department of Health Research Ethics Committee, as well as the Department of Health, Vhembe District. The researcher provided the participants with information regarding the study to have an idea of what consenting for. To ensure no harm, the researcher asked questions that are unoffending and adopted a non-judgmental attitude toward the participants. Codes were used instead of participants’ actual names, to protect their identity. Information provided by the participants was kept away from people who do not form part of the study.

## 3. Results

The study involved a total of 9 caregivers, of which four were mothers and five were grandmothers. This is illustrated in Table 1.

Most of the participants were grandmothers, more than half at the rate of 55.5%, and less than half (44.5%) were mothers. 

A majority of caregivers were not employed and did not go to school at all. Some depended on social grants to feed. Those that managed to go to school only reach grade 12, as depicted in Table 2 above.

Two themes emerged from data analysis, which include a lack of knowledge regarding nutritious food and a knowledge deficit regarding signs of malnutrition.

### 3.1. Theme 1: Lack of Knowledge Regarding Types of Nutritious Food

Findings revealed that participants lacked knowledge regarding nutritious food. Two subthemes emerged from the theme (lack of knowledge regarding types of nutritious food) namely: breastfeeding and non-nutritious food.

#### 3.1.1. Sub-Theme 1.1 Breastfeeding

Breastfeeding is still a challenge to date. Mothers have various reasons they are not breastfeeding exclusively namely job/work and insufficient breast milk.

In this paper, findings revealed that caregivers experienced challenges when it comes to breastfeeding. Mothers were unable to breastfeed their children during the day due to work or school. Hence, they only breastfed at night. Children were not breastfed on demand because the mother was tired from schoolwork or work and wakes up early in the morning to go to work and come back late. This is illustrated in the expression below:

“*The child is only breastfed at night, since the mother is at school during the day and is a grade 12 student who is studying at night until morning*” (Grandmother, 61 years old)

Another caregiver also revealed that children are not exclusively breastfed on demand

“*The mother breastfeeds the child at night only, since the mother is working during the day. The child is fed formula milk during the day, which she stopped at 7 months, so alternatively, gave her boiled ultraMelk but also along the way also refused it*”(Grandmother, 58 years old)

On the other hand, grandmothers encouraged the mothers to stop breastfeeding the child because the child is not eating food. Their perception is that breast milk is not more important than food. The quotes below show the reason she stopped breastfeeding her child.

“*Yes, I was breastfeeding him exclusively but then started giving him soft porridge* (prepared by mixing maize meal and water only) *at around 3 or 4 months if not mistaken because my grandmother feels the child is not satisfied of breast milk “mikando” only. Even so, the child chose breast milk over soft porridge so my grandmother decided that we stop giving breast milk at 7 months, she believed that if I stop breastfeeding him, he will be so hungry he will eat soft porridge*”(Mother, 33 years old)

#### 3.1.2. Sub-Theme 1.2 Non-Nutritious Food

Most participants have poor food choices, but these food choices of primary caregivers (mothers) were different from the food choices of secondary caregivers (grandmothers). 

Mothers were found to be feeding their children junk food, e.g., Danone yoghurt, sweets, puree, and juice for kids, because they felt that it is more nutritious than feeding the child soft porridge. The following quotes depict their poor food choices. Participants between 20 and 29 years of age fed their children junk food

“*The child is not fed soft porridge only; but I also give her Danone yoghurt, sweets, and juice for children*”(Mother, 21 years old)

“*She really enjoys potatoes; I mean potato chips, porridge with artchaar, but Danone yoghurt and juice for children is her most favorable food*”(Mother, 20)

Grandmothers feed the children soft porridge in the morning, and although they tried to provide a variety of things like Cremora powdered coffee creamer, sugar, etc. it was not nutritious enough.

“*But I don’t give her soft porridge* (mix maize meal into a boiling water in a pot and stir until its thick) *but add with some sugar only; I sometimes add “Cremora” powdered coffee creamer to soft porridge since I can’t afford to buy formula. But I think I’m feeding him quite right since I’m not giving him soft porridge with sugar only but with Cremora powdered coffee creamer*”(Grandmother, 49 years old)

“*I give the child soft porridge every morning, then pap with vegetables, sometimes with mango artchaar*” (Grandmother, 59 years old)

### 3.2. Theme 2: Knowledge Deficit Regarding Signs of Malnutrition

Participants also reflected having difficulties in identifying a child with malnutrition. This was identified through a child’s physical appearance.

#### Sub-Theme 2.1 Physical Appearance

Recognizing a child with signs and symptoms of malnutrition is difficult. In this paper, few participants referred to the child with malnutrition as healthy but having other conditions like diarrhea. This statement was supported by the following quotation:

“*Nurse, I’m surprised that you are saying my grandchild is having Kwashiorkor, really its confusing me, because I can see she is healthy “shining” and fat, so where does this Kwash come from? My grandchild is only passing loose stools and “Ngoma” (fontanel) which is not closing but getting bigger*”(Grandmother, 66 years old)

Inability to identify malnutrition in the child leads to improper treatment. Hence, the caregiver went to a traditional healer to seek help as she believed the child was having diarrhea, which is caused by a “Ngoma” fontanel that is too big and not closing.

“*I also took the child to traditional healer, because the traditional healer knows best about “Ngoma” fontanel that is not closing, she is her traditional healer after all. She will know what to do to help this child since” Ngoma” fontanel is not closing; it is getting bigger, so when I arrive, she gave me some “muthi” traditional medicine to apply on the head*”(Grandmother, 69 year old)

She went to the nearest clinic for treatment of diarrhea, because she understands it is a better place to be treated at. That’s when she was told about malnutrition which she was not aware of. She used treatment from the clinic and that from a traditional healer. The quotes show how sure she is that both western and traditional medicine work.

“*Ok does this means you don’t know that your western medications are made from the trees found in the bush, it is just that this white people cooked it, in Tshivenda the traditional healer, burns and pounds them to make “muuluso”, a pounded traditional medicine used to protect the child from illness, haven’t you heard about “muuluso*”(Grandmother, 69 years old)

## 4. Discussion

The study findings revealed that caregivers experienced challenges when it comes to breastfeeding. Mothers were unable to breastfeed their children during the day due to work or school, they only breastfed at night. Prashanth et al.; Pravana et al. [18,19] indicated that 42.7% of children were not exclusively fed breast milk for up to six months. Furthermore, it was revealed that children who started with complementary foods before six months, and too late, are likely to develop severe acute malnutrition. However, there are still mothers who know the benefits of breastfeeding and practice it. The study by Ansuya et al. [20] found that out of 58.9% of mothers who practiced exclusive breastfeeding, only 1.89% were at risk of being malnourished.

In this study, mothers between the age of 20 and 29 years fed their children Danone yoghurt, sweets and juice for children. It has resulted in a high rate of children who are underfed among the needy population because food is expensive and they can’t afford it [21]. Studies in Karnataka and in Nepal, by Prashanth et al. and Pravana et al. [18,19] show that severe acute malnutrition is common in children born to mothers who are below 20 years. Furthermore, severe acute malnutrition is 3.21 times higher in children born to mothers below 20 years and ≥35 years compared to those mothers aged between 20–34 years. This statement support Table 1, which indicates quite a few children with malnutrition are under the care of caregivers ≥ 35 years of age. This is supported by Drammeh et al. [22] who found that children under the care of teenage mothers or those younger generations are more likely to be undernourished.

While young mothers were giving junk food, grandmothers fed the children soft porridge [23]. Mweemba [23] revealed that not giving enough food, giving food with no nutritious value, and a lack of variety in diet are contributory factors to malnutrition.

Few participants referred to their malnourished children as healthy but having other conditions such as diarrhea. Inability to identify malnutrition in the child leads to improper treatment. In one case, the caregiver went to a traditional healer to seek help as she believed the child was having diarrhea, which is caused by a big fontanel. Mweemba [23] conducted a study in Lusaka and they also confirmed that symptoms of kwashiorkor were confused with other illnesses such as diarrhea. Participant confessed that kwashiorkor is difficult to assess as the child looked fat. This is in line with the findings of this study as caregivers referred a child’s unhealthy body to a healthy one. Similarly, Mohseni et al. [12] indicated that knowledge mothers have about their children’s health has a significant impact on the children’s nutrition level.

## 5. Conclusions

The study findings describe the perceptions of caregivers regarding malnutrition. Caregivers perceived that the initiation of complementary food early, and not breastfeeding exclusively, were the factors that contributed most to malnutrition. This study will give detail to policy developers and other concerned bodies to combat malnutrition by coming up with appropriate nutritional educational programs including breastfeeding which will reduce the child mortality rate. Furthermore, health care workers could then provide relevant health education which will help in preventing malnutrition and promoting growth.

## Figures and Tables

**Table 1 children-09-01784-t001:** Age of Caregivers.

Age of Caregivers	Number of Caregivers	Relationship to the Child
20–29 years	02	Mothers
30–39 years	01	Mother
40–49 years	01	Mother
50–59 years	02	Grandmothers
60–69 years	03	Grandmothers

**Table 2 children-09-01784-t002:** Age, employment status and educational level of caregivers.

Age of Caregivers		Educational Level
20 Years	Unemployed	Grade 12
21 Years	Unemployed	Grade 8
33 Years	Working at the farm	Grade 11
49 Years	Self employed	Grade 2
58 Years	Unemployed	Not educated
59 years	Unemployed	Not educated
61 Years	Pensioner	Not educated
66 Years	Pensioner	Not educated
66 Years	Pensioner	Not educated

## Data Availability

Data sharing will be made available to other researchers on request within the regulations of the University of Venda.

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
