# Peer review of "Perceptions of Caregivers Regarding Malnutrition in Children under Five in Rural Areas, South Africa†"

_children, 2022, doi:10.3390/children9111784_

Round 1

Reviewer 1 Report

Dear Author

Please change the Malnutrition in the title to non nutritious foods. 

And also in the manuscript.

Author Response

The term malnutrition was replaced with non-nutritious foods in the title and in the manuscript

Reviewer 2 Report

Dear Authors,

Thank you for sharing your study. It is interesting to learn about the malnutrition issue in your country. However, it will be more beneficial or clearer to the readers if the local terms used can be elaborated further, e.g. Cremora, purity, ehee, Kwash, "Ngoma" etc?

For soft porridge, how does this dish prepare in your country?

Is Danone a food company name or a local dish? 

References: Do check the in-text citation for the whole manuscript. For an example, it is more appropriate to state ..according to Children's Act 2005 [1], and not according to [1].

Introduction: Para 1, l. 34m .. and soya foods Fish, chicken...- Do check the sentence again. 

Para 3, Malnutrition - no need to be capitalised. 

Methods: suggest using a formal language:

Para.1, l.71, "dig deep" - to be removed perhaps? 

Para 4, l.90, "picked" - selected?? 

Suggest using passive voice and complete sentence: 

Para 4, l. 90, Then wrote the names of the caregivers.. Revise to "The names of the caregivers were then written..."

Results: All caregivers were female, and included primary and secondary caregivers. What are the socio-economic status and education level of the caregivers? As the sample size is small (N=9), good to stay consistent in reporting the number of participants, rather than %.

Table 1 reflects the age distribution of the caregivers, but no result on the types of food they give to the children or malnutrition status, which has been cited in the manuscript, e.g. para 1 of results, l.150, para 2 of subtheme 1.2, l. 192, and para 2 of Discussion, l.248, respectively. Do revisit the Table 1. 

Discussions: Para.1, l. 240, the value 1.89 - do specify the unit. 

Para 2: Less than half of the participants were mother, and it is not clear the age of mother and caregiver. It is not strong to support the statement the younger mother are more likely to be undernourished from the present study.

Suggest including the study limitations and strength in the discussion

Conclusions: L.268, the findings did not indicate caregiver's perception that caregiver's age is one factor contributing malnutrition.

Be specific on what information can be translated to the policy developers in combating malnutrition from your present qualitative study.

Author Response

  1. Local terms like Ngoma  were elaborated
  2. Preparation for soft porridge explained
  3. In-text references checked
  4. The term malnutrition was written in lowercase
  5. The word "dig deep" was removed
  6. The word "picked" was replaced with "selected"
  7. Line 104 is rephrased to "the names of the caregivers were then written"
  8. The socioeconomic and educational status of the participants were included
  9. The table was revisited for additional information
  10. Food given to children by young mothers and grandmothers indicated
  11. Line 318 to 322 discussed caregiver's perceptions and contributing factor to malnutrition

Reviewer 3 Report

This paper sought to highlight the perceptions of caregivers on malnutrition in rural South Africa. Overall I believe it is important to have a subjective approach to malnutrition especially from the caregivers perspective so as to compare this with the objective approach that is used in almost all studies. However, I believe the paper needs to be improved to better highlight the issue.

Introduction: The introduction does not really explain why research was necessary. The last paragraph attempted to do this by stating a few studies that have been done on malnutrition and stated that none had been done on caregivers. At this point the researchers needed to state why this information was needed and what it would add to existing literature.

Also throughout the paper, some references should be spelled out instead of just having the number. For example references in the last paragraph (lines 54-66) makes the paragraph hard to comprehend.

Methods: The authors need to be clear on the sampling method. State clearly when purposive sampling was used and where random sampling was used. Also, the justfication for using purposive sampling does not seem appropriate... "Sampling techniques were purposive because caregivers are the ones taking care of 85 the children with malnutrition"... that does not justify the sampling method.

The authors could add a few sentences on why they considered using unstructured interview verses using structured interview for their data collection.

Results: the first paragraph needs to be rewritten to describe what is seen on table 1. With only 9 participants, the sentence ..."Participants’ age in this study ranges from 20 to 69 years and the age of caregivers determined the type of food they give to the children"... is not accurate. Also, this information is not reflected on the table as the authors state.

Rewrite the sentence lie 155-156 to remove repetition present on the sentence.

Authors need to refrain from stating that age determined type of food given because this was a qualitative study and not quantitative. Instead they should just describe the observation.

Some of the foods mentioned are not globally recognized and the authors could give a brief description of what they are. They have done this for some but not all.

Discussion. Overall the discussion needs to be redone. the study is on caregivers perception on malnutrition but the discussion focused more on feeding practices and maternal age.

I do not think the results provided enough information for this conclusion .."In this study, mothers between the age of 20-29 years fed their children junk food due  to a lack of maturity and don’t have enough time to prepare food.."

Conclusion: ..."Caregivers perceived that caregiver’s age, initiation of complementary food early, 270 not breastfeeding exclusively, and on-demand were the most factors that contributed to 271 malnutrition".... This was the perception of the authors not caregivers according to the results.

This section needs to be revised too as the other sections are revised. For this study, I believe the results probably provide basis for other studies that can inform policy and nutrition education efforts but the results here are not descriptive or detailed enough to inform policy by themselves.

References: These need editing and proper formatting. At the very least, the authors should have removed the extra numbers at the beginning of the references after they automatically numbered the references.

Author Response

  1. The importance of conducting the study was indicated in lines 72 to 73
  2. References were spelled out
  3. The rationale for choosing purposive sampling was explained. in lines 97 and 98
  4. Explanations on the use of unstructured interviews over structured ones were done
  5. Description of the table was done
  6. Revisions done on the discussion part
  7. Conclusions attended to

Round 2

Reviewer 1 Report

Hi,

Thank you for considering the comments-

Author Response

Thank you for your words of appreciation

Reviewer 3 Report

I appreciate that the authors tried to incorporate all the changes suggested. However, when incorporating the changes, the overall study became confusing. For example, the title of the study was changed to perceptions of caregivers on non nutritious foods, previously it was perception on malnutrition. If the study title and purpose changed, then the key question of the interviews would not derive the information that was being sought. According to the authors, the question for the unstructured interviews was "can you please tell me what you think may be the cause of malnutrition in the child?" With this question the original title and purpose of the study was more fitting. Instead of changing the title, the authors could have improved the content to include more studies on malnutrition in children and provide some review on studies done on malnutrition in that region and country. Also, with the current title, the lines 58-68 of the introduction should have included a review of what has been done on infant feeding practices specifically in South Africa and if possible in the area of study instead of having a collection of studies that have been done on child malnutrition all over the world. The last statement (line 72-73) that is meant to describe the purpose of the study still states that the purpose was to investigate the perceptions on malnutrition.

In the methods section the authors still mentioned that the caregivers were purposively selected in one part (line 97-98) and randomly selected in another (line 103).

In the results, the description of the table is "Age of caregivers" which is not what the table shows. There is more than age of the caregivers

Table 2 is not even mentioned on the text

I believe that the quotes should not be edited to be more descriptive. Instead, the description should be done in the text explaining the quotes. If the interview was done in another language then interpreted then that can be explained but the quotes should not be edited.

The discussion and conclusion could also be improved considering that the title and purpose of the study was changed.  I still think with the number of participants and the information provided, the authors would not be able to conclude that young caregivers fed their children junk food more than older caregivers.

Author Response

Thank you for the valuable comments. Kindly note that the change in the topic from malnutrition to non-nutritious foods was done to address one of the reviewers' comments, which was also confusing to us. However, the topic will be changed to its original draft to suit the content. 

Table 2 was corrected and was included in the text